



# Atmospheric stratification over the southeast Atlantic Ocean adjacent to the Namibian coast

Abdulaziz T. Yakubu[1], Danitza Klopper[1, 2], Henno Havenga[1], Roelof Burger[1], Paola Formenti[3], and Stuart J. Piketh[1]

[1]North-West University, School for Geo- and Spatial Sciences, Potchefstroom, South Africa
[2]University of Limpopo, Department of Geography and Environmental Studies, Polokwane, South Africa
[3]Université Paris Cité and Univ Paris Est Creteil, CNRS, LISA, F-75013 Paris, France

**Correspondence:** Stuart J. Piketh (stuart.piketh@nwu.ac.za)

**Abstract.** Atmospheric stability, characterised by absolutely stable atmosphere and temperature inversions, is a prominent feature of atmospheric stratification over the Namibian coast and the adjacent Southeast Atlantic (SEA) Ocean. This stratification plays a critical role in the transport and distribution of atmospheric pollutants, energy, momentum, and other physical components. Key factors influencing the vertical structure of the atmosphere include the South Atlantic High and transient
5 baroclinic westerlies, which drive ocean-subcontinent heat exchange, a dominant factor in the formation of inversions and atmospheric stability. Observations from ERA5 reanalysis data and global positioning system radio occultation (GPS-RO) measurements over 11 years (2007–2017) reveal that low-level inversions are more frequent over the Namibian coastal regions and adjacent oceans compared to the subcontinent. This predominance is attributed to higher atmospheric stability, linked to ocean-atmosphere heat interactions driven by the cold Benguela western boundary current. Surface inversions exhibit a sea-
10 sonal pattern, mostly peaking during winter. Moreover, winter is associated with a generally lower inversion base height ($h_{ib}$), while summer favours elevated-based (EL) inversions. Interestingly, inversions are strongest during the last part of winter and weakest in autumn, with typical depths ranging from 10 to 125 hPa and strengths between 3 °C and 9 °C. These inversions exhibit a significant association with the conditions that may support the formation of stratocumulus clouds, often occurring nearer to the surface within the planetary boundary layer (PBL).

## 1 Introduction

The west coast of Southern Africa and the Southeast Atlantic (SEA) Ocean serves as a natural atmospheric laboratory, characterised by unique meteorological regimes and the influence of regional aerosol-cloud-radiation interactions (Tyson and Preston-Whyte, 2014; Fuchs et al., 2017). Geographically positioned in the subtropics and mid-latitudes, Namibia's mean atmospheric circulation is governed by the high-pressure belt associated with the poleward descending of the Hadley cell (Dyson and
20 Van Heerden, 2002; Goudie and Viles, 2015). In this region, large-scale air subsidence in the lower troposphere leads to adiabatic heating of the upper layers, stabilising and often decoupling the lower tropospheric region and boundary layer to create stable stratified layers (Wood and Bretherton, 2004; Zhang et al., 2010).





Over the SEA Ocean, stability is further enhanced by mesoscale processes, such as throughout the year cool sea surface temperatures (SSTs) associated with the adjacent coastal upwelling around the Namibian coast (Zhang et al., 2010; Painemal et al., 2015). This stability is reinforced by the semi-permanent South Atlantic and continental high-pressure systems (Taljaard, 1995; Celia et al., 2004; Tyson and Preston-Whyte, 2014; Gordon et al., 2018). These conditions foster the formation and persistence of diurnally changing high-speed coastal low-level jets (LLJs) like the Benguela LLJ (Nicholson, 2010; Lima et al., 2019). Atmospheric stability, characterised by decreasing environmental lapse rate compared to the adiabatic lapse rate (Stone and Carlson, 1979; Arya, 1988; Cosijn and Tyson, 1996; Warner, 2004), contributes to conditions that can support fog formation, especially when combined with factors such as high moisture availability and cooling of the surface air. These stable conditions help trap moist air near the surface, which, when cooled sufficiently, can lead to frequent fog events (Seely and Henschel, 1998; Kaseke et al., 2018). Over the Namibian coast and adjacent ocean, high oceanic moisture and vertically inhibited cloud development above the coastal and marine boundary layers promote a semi-permanent stratocumulus (Sc) cloud deck formation (Wood, 2012, 2015), spanning 10°–30° S and 10° W–10° E (Muhlbauer et al., 2014).

Notably, higher low-level cloud cover correlates with increased lower atmospheric stability during the biomass-burning season (Ryoo et al., 2021). However, the interactions between different aerosol regimes and their impacts on low-level cloud formation under stable atmospheric conditions remain unclear.

Since the late 1990s, large-scale field campaigns have explored the transport of biomass-burning aerosols (BBA) emitted from seasonal forest fires in Central Africa (Lindesay and Tyson, 1990; Swap et al., 2003; Zuidema et al., 2016; Formenti et al., 2019; Haywood et al., 2021; Redemann et al., 2021). Dense and persistent BBA plumes are transported within stratified layers at various altitudes (Adebiyi and Zuidema, 2016; Diamond et al., 2018; Gordon et al., 2018). Similarly, pollution from industrial regions like South Africa's Highveld is redistributed in optically thin stable layers within the free troposphere (Garstang et al., 1996; Tyson et al., 1996; Freiman and Piketh, 2003). While light-absorbing aerosols may be transported within the Namibian marine boundary layer (Formenti et al., 2018), the strong thermal inversion above the stratocumulus cloud deck often inhibits vertical exchanges between the BBA layer and clouds.

Given these dynamics, understanding the characteristics of inversion strength and its influence on pollutants, including aeolian particulates, becomes critical. Particle-free air gaps, observed above, below, and close to pollution plumes and clouds, vary in depth and location (Haywood et al., 2004; Wilcox, 2010; Costantino and Bréon, 2013; Dagan et al., 2016; LeBlanc et al., 2020; Pennypacker et al., 2020). These gaps can significantly impact cloud properties, but their variability requires further investigation. Moreover, vertical atmospheric stability contributes to the persistence of lofted, light-absorbing aerosol plumes, which influence the regional radiative budget over large spatial and temporal scales (Mallet et al., 2019; Flamant et al., 2022).

Efforts to study the relationship between elevated inversions and aerosol transport over Namibia's west coast have primarily relied on radiosonde balloons launched either daily over extended periods or multiple times on a short-term basis during field campaigns (e.g., Preston-Whyte et al., 1977; Cosijn and Tyson, 1996; Swap et al., 1996; Formenti et al., 2019; Haywood et al., 2021; Zuidema et al., 2018; Redemann et al., 2021). Modelling studies (Painemal et al., 2015; Mallet et al., 2019) have also been employed to explore these relationships. However, the limited spatiotemporal coverage of radiosonde data and uncertainties in model simulations present challenges. Over the past two decades, global positioning system radio occultation





(GPS-RO) has emerged as a reliable alternative, offering extensive spatial and temporal coverage (Shyam, 2019; Xie et al., 2006, 2012; Ao et al., 2012; Alexander et al., 2014).

This study uses an 11-year climatology (2007–2017) of thermal atmospheric stratification below 10 km above ground level (AGL) over Namibia and the adjacent Atlantic Ocean, combining ERA5 reanalysis model data with GPS-RO observations from the COSMIC mission to evaluate the regional atmospheric stratification's spatial and temporal variability. It seeks to understand the impact of stratus clouds on the formation of the stratocumulus cloud deck and to investigate the relationship between the strength of atmospheric stability, as identified by the data, and the lower tropospheric stability (LTS), which has been shown

to correlate with cloud fraction in the region. The research also assesses the interaction between atmospheric stratification and different aerosol regimes over both the land and the adjacent ocean. It also examines how the position of the land's lowest stable atmospheric layer influences the height of biomass-burning aerosols transported off the West Coast.

## 2   Data and Methods

### 2.1   Region of investigation

This study focuses on the boundaries defined by 15°–30° S and 0°–20° E along the west coast of Namibia (Fig. 1). The area has been divided into three regions, namely, a coastal margin (outlined in blue in Fig. 1), the ocean, and the land (over Namibia). The coastal margin was demarcated as a 5° band along the coast just offshore and over the low-elevation regions below the continental plateau, as seen in Fig. 1. This region along the coastal desert represents a transition zone between the cold Benguela current and SEA Ocean and arid Namibia (Preston-Whyte et al., 1977; Cosijn and Tyson, 1996; Garstang et al.,

1996; Tyson and D'Abreton, 1998; Ao et al., 2012).

### 2.2   Data collection and processing

The fifth-generation European Centre for Medium-Range Weather Forecasts (ECMWF) Reanalysis (ERA5) ensemble reanalysis model data is a robust record of the global atmosphere, land, and ocean that spans a period from 1940 onward (Hersbach et al., 2023). Furthermore, the model uses a 10 member ensemble at three-hour intervals to estimate the associated uncertainty.

This work mainly utilises the regular 0.25° X 0.25° gridded monthly data at 3-time steps (i.e., 00:00, 12:00, and 18:00 UTC) for the variables: air temperature at pressure height between 100 to 1000 hPa (Tx °C, where $100 \leq x \leq 1000$) and cloud cover fraction (CF) between 100 and 1000 hPa.

In addition to the ERA5 reanalysis, the study uses the GPS-RO observation data. The Global Positioning System in low Earth orbit (GPS-LEO) aboard the COSMIC satellite launched in 2006 delivers over 500 profiles of data on daily limb-soundings of

the global atmosphere (Kursinski et al., 1997; Anthes et al., 2008). GPS data has been demonstrated to be a suitable substitute for radiosondes across the globe by promising expanded data coverage, more frequent timespans and widespread spatial scales (Guo et al., 2011; Alexander et al., 2014; Hande et al., 2015). The atmospheric refractivity (N) vertical profiles contained in the publicly available atmPrf and the corrected version wetPrf datasets are extracted from the raw radio signals through



a Radio Occultation (RO) inversion procedure by the COSMIC Data Analysis and Archive Centre (CDAAC) software. The
inversion scheme follows the initial retrieval of the bending angles, from which the N profiles are obtained (Kuo et al., 2004).
Subsequently, the sounding parameters are retrieved from the N profiles based on the principle of radio refractive index in air,
a procedure described in various work (e.g., Bean and Dutton, 1968; Kursinski et al., 1997). Assuming a neutral atmosphere, N
can be related to pressure (P), air temperature (T), and vapour pressure ($P_w$) by the expression (Guo et al., 2011; Hande et al.,
2015)

$$N = 77.6P/T + 3.73*10^5 Pw/T^2, \tag{1}$$

such that the parameters retain the standard units of hPa and Kelvin (i.e., for pressures and temperature, respectively). The
bending angles of GPS-RO N profiles in Eq. (1) are proportional to atmospheric moisture with associated uncertainties for each
profile (von Engeln and Teixeira, 2013; Ratnam and Basha, 2010; Guo et al., 2011; Wang et al., 2013). These uncertainties are
corrected with a 1-D variational analysis product from the ECMWF (Guo et al., 2011; Shyam, 2019). The resulting data has an
interpolated vertical resolution of 100 m (Hande et al., 2015). Furthermore, data for the atmospheric structure were excluded
if the refractivity signal was superrefracted, i.e. where the critical value of -157 N km$^{-1}$ was exceeded (Sokolovskiy, 2003;
Guo et al., 2011). A total of 1841 out of 33023 profiles contained superrefracted signals. Hence, they were not considered in
the analysis for this work (1619 out of 19937 over the ocean, 198 out of 6561 over the coastal margin, and 24 out of 6525
over land). Between 2007 and 2017, 60 % of the profiles were usable from 100 m agl and another 20 % from 200 m agl.
Superrefraction appeared most frequently at $600 \pm 200$ m agl. The spatial and temporal distribution of valid profiles across the
study area is given in Fig. 1. Months with less than ten measurements per region were also excluded from further analysis.

Temperature inversion layers are estimated as the top of a layer of increasing temperature with height. Temperature inversion
strength, describing the local stability of the atmosphere as the change in temperature per 100 m interval, and inversion depth
as the vertical height through which the inversion persists, are also calculated. Shallow isothermal layers directly above or
below the inversion are not included in the calculation of the inversion depth. Also, the cloud-aerosol-transport system (CATS)
observation data is used to track aerosol transport at different altitudes over the region. The CATS instrument on board the
International Space Station (ISS) was launched in 2015 and operated for approximately 3-years, observing the properties of
aerosols and clouds. CATS mainly comprise a lidar system and photon counting detector, thus deriving cloud and aerosol
layer properties such as backscatter, depolarisation ratio, and layer height at wavelengths 532 and 1064 nm (McGill et al.,
2015; Proestakis et al., 2019). This work further uses the 5 km horizontal resolution CATS Level 2 version-3 aerosol layer
product to examine the influence of inversion on air pollutants. The layer product, primarily derived from CATS Level 1B data,
provides information on the vertical distribution of aerosol and cloud. Through algorithms similar to Cloud-Aerosol Lidar and
Infrared Pathfinder Satellite Observation (CALIPSO) cloud–aerosol discrimination (CAD), the product provides parameter
based datasets on identified atmospheric layers to include feature type (i.e., aerosol or cloud) and subtypes (Proestakis et al.,
2019). The classification scheme identifies eight aerosol subtypes: dust, dust mixture, smoke, marine, marine mixture, polluted





continental, clean/background, and volcanic, which are used to define the aerosol types found at different vertical levels in the work. Further information regarding the CATS instrument and the theoretical basis is available on (Proestakis et al., 2019).

## 3    Result and discussion

### 3.1    Comparison of ERA5 model and GPS-RO data in observing the vertical profile

This section compares the ERA5 model inversion profile to the output from the GPS observation data, which includes assessing the inversion depth and strength. Hence, Fig. 2 illustrates the validation of the ERA5 monthly average against the GPS dataset. Only the locations within longitudes 8, 10, and 13° E were available for the GPS plots during this study. Therefore, the corresponding ERA5 data within this boundary were compared based on point locations. Invariably, the data presented here represents mainly areas around the coastal boundary towards the SEA Ocean. From the plots, the ERA5 temperature profile

demonstrates strong agreement with the GPS observation during the study period. However, the modelled data slightly differs at the point of inversion detection, resulting in varying values of the inversion depth and strength between it and the GPS, especially for EL inversion. Nonetheless, and in most cases, the inversion base height ($h_{ib}$) is formed at about the same altitude in both SB and EL inversion datasets. Notably, most GPS measurements do not cover heights below 500 m above ground, which could affect the effective determination of SB inversion. As stated earlier in the previous section under data description,

only data points above this height are considered for GPS analysis. In contrast, profiles of heights below 500 m are constantly shown for ERA5. Irrespective of these discrepancies, the temperature profiles ERA5 Vs GPS demonstrated good agreement with strong correlations (i.e., R > 0.95) across the study period. Also, the uncertainty in the ERA5 modelled data relative to the GPS observation is ±0.014, which is considerably minimal. Nevertheless, it is essential to point out that the uncertainty at low levels, especially during the summer months, is significantly higher (i.e., ±0.11 and ±0.17, respectively) than at high altitude

and during winter. This wide gap of uncertainties accounts for the observable differences in both datasets' inversion depth and strength values. Furthermore, both datasets demonstrated the association of SB inversion with the autumn/winter season and EL inversion with the spring/summer period. The minimum inversion depth from the GPS occurred during the autumn and the maximum in the winter. As earlier seen with the ERA5 data, the min/max aligns well in both datasets. Similarly, the inversion strength reaches a minimum during autumn for the datasets and a maximum in winter and spring respectively for GPS and

ERA5.

### 3.2    Variability in atmospheric stratification over Namibia

To describe the variability in the stratified layer of the lower atmosphere over the Southeast Atlantic (SEA) Ocean and the coast of Namibia, Fig. 3 illustrates the ERA5 monthly mean temperature profile between 2007 and 2017 at 00:00 UTC for the region. Inversion formation over the region appears in distinct levels, which are grouped in this context into three main levels:

high-level (200-400 hPa), mid-level (450-750 hPa) and low-level (800-1000 hPa) inversions. While the high and mid-level inversions occur less frequently or are not commonly detected, the low-level inversion tends to be the most common. From the





figure, low-level inversion occurs more obviously over the ocean than on land. This inclination is associated with the South Atlantic high-pressure system being the primary driver of subsidence, causing the inversion, hence having its core over the ocean (Tyson and Preston-Whyte, 2014). Besides the baroclinity resulting from the South Atlantic high (SAh), interaction with the low continental easterly advection also plays a vital role. Thus, the inversion tends to form over the region throughout the months and seasons and significantly varies in frequency, base height, strength, and depth at different periods.

Two patterns of inversion are observed to be formed over the region, including the inversion formed on or close to the surface (i.e., between 0 and 50 m; surface-based inversion SB) and the ones formed some height above the surface (i.e., > 50 m and above; elevated inversion EL). From observation, only the EL inversion pattern is observed across the continental part of the study region over the year. However, the coastal areas are split into two patterns depending on the seasons. While the autumn and winter seasons are associated with the SB inversion, the spring and summer months are dominated by EL inversion. This characteristic seasonal variation in the inversion base height ($h_{ib}$) is consistent with existing findings (Tyson and Preston-Whyte, 2014). Variation in the formation pattern or point of initiation of the inversion is a feature linked to the change in the pressure magnitude between winter and summer. Typically, weak South Atlantic anticyclones over the ocean coupled with a strong Benguela current (i.e., through modification and stabilisation of the lower troposphere) amid lower surface heating over the land during winter result in SB inversion. Meanwhile, EL inversion mainly occurs during summer due to a generally strong South Atlantic high-pressure system, which causes sinking air to warm adiabatically until the temperature is higher than the air aloft (i.e., the air in the mixed boundary layer). Of the leading causes of inversion formation, similar variation is observed in North America along the California coast, which is consistent with the findings herein (see Burk and Thompson, 1996).

Beyond the inversion formation initiating from the surface or at an elevation, depending on the season, the base height of the inversion significantly varies monthly. Thus, the base height often increases during spring through summer, with the peak in January (i.e., $h_{ib} \approx 250$ m), while $h_{ib}$ mainly decreases through the autumn to winter months and is less than 50 m in elevation. A similar variation is observed through the latitudinal direction such that the base height decreases when moving from the Namibia coast poleward in the southern hemisphere. Meanwhile, $h_{ib}$ variation over the land and ocean relative to the Namibian coast is considerably similar. Moving away from the coast inwards, the continental or oceanic environment follows a significant increase in the base height. Hence, $h_{ib}$ is minimum around the coast and increases moving away in either direction (i.e., towards the land or ocean). Suggestively, this variation is liable to the height of advection of cool ocean air over the warm continental at the coastal boundary during inversion formation (Tyson and Preston-Whyte, 2014; Oke, 1978).

Figure 4 illustrates the monthly average invasion occurrence frequency between the surface and height 950 hPa. Spatially, the inversion forms more frequently in the atmosphere over the ocean and the coastal boundary of Namibia than in the inland areas. Besides, the frequency tends to be higher around the coast than in areas away from the coastal boundary. The frequency mainly peaks around 15-25° S along the Namibian coastal border (12-15° E), with the highest occurrence around 24° S. Significantly, the variation is consistently associated with all periods (i.e., all months and seasons). Temporally, the inversion occurrence frequency increases from summer to autumn, after which it decreases from winter back to summer. The peak is registered during May (i.e., frequency > 85), while the lowest occurred in January (20 < frequency ≤ 40). Variation in the frequency is mainly influenced by the differences in the changing rate of land surface temperature compared to the ocean due to heating from





solar radiation. Also, differences in cloud cover (or cloud fraction CF) over the two environments are critical in influencing this variability. As is well known, the Southeast Atlantic Ocean is often covered by stratocumulus decks (Sc), which mainly reflects solar radiation back to space, causing cooling effects on the surface. Subsequently, inversion is formed due to subsidence resulting from the slight temperature differences between the cool and warm air at the coastal-continental boundary. Here,

cooler air from the Benguela current and the ocean upwelling tends to force the warmer air from the subcontinent to lift above it. Noticeably from the figure, the inversion over the land between the surface and 950 hPa is uncommon. Often, subsidence occurs over the subcontinent part of Namibia at a pressure height between 850 and 800 hPa above the surface level. Moreover, the low-level inversion over the inland is mainly in the order of frequency less than 20 and common in all months and seasons.

Figure 5 shows a more pronounced variation over the ocean than the land for the inversion strength and depth. Above the ocean, the inversion strength decreases during the summer and autumn seasons, with the lowest experienced in March, and increases during winter and spring, with the peak occurring in September. Like observed across the maritime environs, stronger inversion is often linked to winter as observed over the land. The differences in the rate of surface heating and radiative cooling during the summer/autumn and winter/spring periods could be liable for this pattern. Typically, the inversion strength ranges

between 3-5 °C in summer/autumn and can reach ≈ 9 °C in the winter and spring, depending on the environment (i.e., land or ocean). The characteristic high inversion strength in winter/spring is a function of the South Atlantic high-pressure system's intense pressure gradient over the SEA Ocean. A common trend in the inversion strength follows the shift from highest to lowest, moving from over the ocean through the coast to the inland, and is associated with subsidence over the SEA and increased outflow in the anticyclone circulation over land (Piketh et al., 1999). Also, cold air near the ocean's surface often

promotes atmospheric stability, especially at low levels, and increases subsidence. In the presence of a high-pressure system such as the South Atlantic high, colder air sinks slowly, thus creating a warmer layer aloft the lower colder air, resulting in an inversion. Further to the changes in inversion strength spatially, along the meridional, the inversion intensity decreases poleward to the south and increases equatorward. On average, the strength reduces by ≈ 1 °C for every 2° poleward south owing to the generally low temperature and reduced solar radiation associated with the southern hemisphere environment. In

this zone, low surface heating by solar radiation reaching it is prevalent in the warming and advection process that yields the subsidence, then inversion caused by the juxtaposition of warm air and the cold wind from the Southern Hemisphere polar region.

In terms of the inversion depths, an increase is visible over the ocean compared to the land surface. The extent of the depth can range from a few pressure heights over the land surface to > 75 hPa over the sea surface. Seasonally, the inversion depth

decreases during summer and autumn; the reverse is true in winter/spring . Remarkably, the inversion strength and depth share similar intensity characteristics over the ocean and land, and during the winter/spring. Within the study period, the minimum depth (0-50 hPa, depending on the geographical location) is recorded in March, while the maximum (10-100 hPa, depending on location) occurs in August. Considering the meridional variations in the inversion depth, there is no significant change in the depth mainly over the ocean except for regions closer to the coast (i.e., location > 12° E), which slightly increase

in depth poleward to the southern hemisphere. Above the inland, the variation in the depth is more noticeable than in the ocean. By comparison, the two environments demonstrates a common increase in depth near the coast. Further characteristic





depth increase is demonstrated above the subcontinent in the meridional direction south. The specific cause of the discrepancy between the environments is yet unclear. However, the similarity displayed along the coastal boundary results from the inversion caused by the advection of warm continental air over the colder ocean air.

Figure 6 illustrates the diurnal variations of the inversion intensity and depth for the period under investigation. Slight changes in depth and strength are noticeable between midnight and evening (i.e., between 00:00 and 18:00 UTC), with an intensive rise in both properties around midnight. Studies have described intense inversion strength and depth over Southern Africa as nocturnal, especially for SB inversion(see Tyson and Preston-Whyte, 2014). This attribute is primarily linked to radiative inversions, such as fog experienced over the Namibian coast, which is formed due to near-surface cooling or elevation

of warm air over colder ones due to the advection effect. Thus, this observation demonstrates good agreement with the earlier stated description. On average, an inversion strength rise of about 2 °C is typically maintained between noon and evening. It can build up to approximately 9 °C at night, depending on the environment (i.e., subcontinent or ocean). Similarly, the depth develops before evening, reaching several hundred meters late at night. Therefore, diurnal variation is mainly influenced by surface heating and radiative cooling during the daytime and night respectively, and other associated factors such as cloud

fraction, relative humidity and terrain, and it possesses the most potent effect at night.

### 3.3    Link between transported aerosol and the stratified atmosphere

The aerosol transport process is assessed using the CATS satellite observation to evaluate the relationship between the stratified layers and air pollutants over the region. According to reports from existing studies on Southern Africa, aerosols and trace gas emissions from the area are noticed to be recirculated back to the subcontinent where the transport is initiated and the

immediate surroundings (Piketh et al., 1999; Chazette et al., 2019; Yakubu and Chetty, 2020). The recirculation process is mainly induced by the anticyclonic fields around the SEA Ocean following the run-through by westerly waves and the passage of cold fronts (Tyson and Preston-Whyte, 2014). Aerosol concentration, constituents and regional transport are seasonally and environmentally dependent. For instance, particulate matter such as sea salt is significant in all seasons and maritime environments. Meanwhile, the biomass-burning substance is substantial in spring/summer over the subcontinent and mostly

dispersed along its pathway. Irrespective of the period and environment, the inversion often influences the dispersion of the suspended aerosols differently, which is liable to affect the radiative forcing effect. From the previous sections, low-level inversion has been demonstrated to be the predominant expression of the atmosphere stratification over the region. Low-level inversion often inhibits the dispersion of suspended aerosol pollutants through increasing atmospheric stability, thereby increasing the concentration of the pollutant over the area. Following this process, the inversion layer forms a canopy aloft the

boundary layer, thus initiating a gap between the stratocumulus deck and the suspended aerosol, which opposes the cloud's growth and often causes a clear sky. Nevertheless, depending on the particle types being transported, they can modify the characteristics of radiative forcing over the region through absorption and scattering.

In Fig. 7, the feature displayed between the collocated inversion event and the stagnated (or transported) pollutants is generally similar for all seasons. During the winter (see Fig. 7a), which is typically characterised by lower $h_{ib}$ or sometimes SB

inversion, fine aerosol such as combustion emissions (e.g., fossil fuel smokes and industrial emissions) and fine mineral or





aeolian dust (MD) are commonly found at the top of the inversion far away from the boundary layer and the low clouds. The stagnated pollutants reside between 1.5 and 5 km (i.e., the pollutant top at 5 km) above the surface. With aeolian dust constituting the predominant aerosol type forming the haze over the region during autumn/winter, anthropogenic aerosols through industrial and domestic activities follow. Meanwhile, a significant amount of marine aerosols is likewise noticeable during

winter. It will be well noticed that biomass burning smoke is negligible relative to the entire constituent making up the aerosol loading. The reason for the lower concentration is that major biomass-burning activities occur outside this period (Tesfaye et al., 2011). During the summer, as shown in Fig. 7b, the haze concentration is mainly found building up beyond the 5km height (i.e., typically between 1.75 and 10 km). It sometimes extends to 15 km altitude, which is different from observation during the winter. Above the inversion top height, the haze formed constitutes a large amount of aeolian dust, and biomass

burning smoke concentration substantially shoots up far beyond the pre-spring/summer period. However, only a slight difference is seen in the contribution of industrial and other anthropogenic-related emissions to the total aerosol loading at this time of the year (Yakubu and Chetty, 2020). Based on the high aerosol optical depth values, several studies have shown that biomass burning significantly adds to aerosol loading from spring to summer (e.g., Yakubu and Chetty, 2022). Another insight is that biomass-burning aerosol will likely gain buoyancy over other particulates due to its constituents warming above the

surrounding air through their absorption properties (Huang et al., 2016). A similar process can be said of industrial emissions such as sulphur dioxide, in which the oxidation rate is enhanced during summer than winter. Lastly, as seen from the previous section, EL inversion often occurs during summer than winter, pushing the inversion top to a greater height. This difference in top height will widen the gap in which the pollutants are trapped above the surface, extend their reach within the boundary layer and keep the haze further away from the low clouds. Previous documentation has also demonstrated that the average

pollutant top over Southern Africa is clearly defined as being between 4-6 km above the surface, and the haze layer top is often greater in summer than in winter (Tyson and Preston-Whyte, 2014).

### 3.4  Interrelation between the vertical structure and stratocumulus deck form over the Atlantic Ocean

Over the Namibian coast and the SEA Ocean, stratocumulus clouds often spread around the lower troposphere and mainly separated from the elevated aerosol layer by inversion formation. Figure 8 illustrates the combined inversion-cloud cover plots

to examine and describe the relationship between the surface inversion and the stratocumulus cloud deck formed over the study region. The figure shows that an inversion is often accompanied by the emergence of stratocumulus clouds as seen by the increase in cloud cover below the 800 hPa pressure height. Furthermore, a prominent distinction is observed between the cloud cover over the low-level inversion dominating coastal areas and the subcontinent with almost no low-level inversion. Cloud cover spreads upward from the inversion base but far below the inversion top height over the coast and outward to the

ocean (i.e., between latitudes 8 and 13° S). However, over the adjacent subcontinent (i.e., between latitudes 16° and 19° S), the absence of an inversion in the lower troposphere results in cloud formation primarily in the upper parts of the troposphere, appearing as high clouds. Thus, cloud cover increases towards the high cloud and diminishes at the low level. It is therefore clear from the variation that the low-level inversion mainly acts in duality in enhancing stratocumulus cloud formation and building a gap between the formed low-level clouds and the aerosol layer aloft the inversion top height. Consequently, the



growth of the supposed clouds formed under this condition is suppressed due to the blockage between the aerosol particles expected to serve as cloud condensation nuclei (CCN) to enhance the cloud growth (Abel et al., 2020; Calderón et al., 2022). Similarly, since aerosol particles emitted in the subcontinent and transported over the Namibian coast and SEA Ocean are often recirculated back to the subcontinent, the relationship posed by the low-level inversion on aerosol transport suggests that such pollutants move over the subcontinent to enhance high cloud growth. Although no study has been conducted to ascertain that

the trapped aerosol above low-level inversions enhances high cloud growth over the subcontinent, it is crucial to establish this to account for the effective termination of these pollutants.

Further observation of the effect of inversion formation on the stratocumulus cloud deck and seasonal influence associated with the different inversion types (e.g., SB or EL inversion) is also noticeable. As shown in previous sections, EL inversions are commonly linked to the spring/summer season, while SB inversions have a higher frequency of occurrence during autum-

n/winter. Similarly, cloud formation during the latter frequently occurs closer to the ground and is sometimes seen as fog, which extends inward to the subcontinent. The advent of fog over the Namibian coast and subcontinent, with a peak from March to August, thereby reducing visibility over the region, is well documented in the literature (see Nagel, 1962) and consistent with the findings in this study.

In addition to the variation in the temperature profile associated with cloud cover, Fig. 9 illustrates the relationship between

the inversion depth, its strength, and the average cloud cover over the region. The figure shows that occasions with negligible inversion strength often result in the absence of cloud cover. However, high inversion strength does not always result in more cloud cover. Similarly, events of no or insignificant inversion depth result in cloud cover absence, and an increase in depth does not necessarily result in more cloud cover. The variation implies that low-level stratocumulus clouds are formed under the influence of low-level inversion, but the inversion does not necessarily dictate their extent. On the other hand, the absence

of inversion will result in no low-level cloud forming; instead, the clouds will be formed at higher altitudes. Meanwhile, the correlation analysis between the depth, strength and cloud fraction demonstrated a moderate relationship (i.e., R= 0.68 and 0.59 for the depth and strength, respectively). Thus, this supports the finding that the inversion plays a significant role in forming the lower cloud layer but does not necessarily influence its size or magnitude at that point.

## 3.5 Discussion

The vertical structure of the atmosphere over Namibia's coast and the adjacent SEA Ocean plays a significant role in the variability of moisture transport, energy, and momentum in the region's atmosphere. The high-pressure system induces subsidence, and the baroclinic environment created along the coastal boundary due to the coinciding cold ocean winds and the heated air from the subcontinent often generates inversions in the lower troposphere. Similarly, the quasi-barotropic region leads to the formation of an inversion over the subcontinent due to easterly low-pressure waves. These atmospheric conditions provide

favourable environments for trapping pollutants in the lower troposphere and forming a stratocumulus cloud deck over the coast and the adjacent SEA Ocean.

Two types of inversions commonly form over the region, each linked to specific environmental and seasonal conditions. SB typically forms over the coast and ocean, predominantly during the autumn and winter. In contrast, elevated inversion





occurs over the subcontinent and the ocean, with a stronger association with the spring and summer seasons. The difference
in inversion types and their preferential timing can largely be attributed to changes in solar radiation reaching the surface and
varying subsidence rates.

During summer, surface heating especially over the land is significantly higher than in winter, as the ocean takes longer to
heat up. The temperature difference between the land and the ocean causes warm air from the subcontinent to drift toward
the coast and ascend over relatively cooler air. Over the coast and extending outward to the ocean, subsidence, driven by
atmospheric stability and the cooler marine surface, causes inversion to form above the surface. In contrast, over the land, cool
air from the ocean replaces warmer air, leading to less atmospheric stability, increased mixing near the surface and relatively
warmer air aloft. This warmer air is typically situated well above the surface, leading to less frequent elevated inversions over
the subcontinent.

In the case of SB during winter, the temperature difference between land and ocean is minimal due to lower surface radiative
heating. The cold conditions over the land promote atmospheric stability near the surface, and the advection of slightly warmer
air from the subcontinent over the cooler coastal and marine surface contributes to the formation of surface-based inversion,
which is typically weak in depth and strength. Results from studies have demonstrated varying characteristics (e.g., frequency,
depth and strength) of SB and EL inversion as it depends on the location and influencing factors (Li et al., 2019; Zeng et al.,
2022). The linking of SB inversion to winter, mostly at nighttime and over the ocean, and EL inversion to summer is consistent
with all studies, including the current one.

The formation and dispersion of stratocumulus clouds over the coast and ocean are strongly linked to the presence of
inversion. When low-level inversions occur under strong subsidence and atmospheric stability, pollutants trapped within and
beneath the inversion serve as cloud condensation nuclei (CCN), aiding cloud formation beneath the inversion layer. It is
important to note that while inversion properties (depth and strength) influence cloud formation beneath the inversion, they do
not dictate cloud size. Instead, the size of the stratocumulus clouds is influenced by the concentration of trapped particulates,
which play a crucial role in cloud development. Where no inversion is present, such as over the land, low-level clouds are often
absent, while mid- and high-level clouds become more prominent. During this period, reduced atmospheric stability and free
convection conditions led to a deeper mixed air layer and increased pollutant dispersion. Further dispersion of these particulates
above the inversion layer influences the formation of mid- and high-level clouds or even results in their recirculation to the
subcontinent. The close links between low-level inversions and stratocumulus clouds formation are well documented in several
other studies (e.g., Adler et al., 2019; Babić et al., 2019; Dione et al., 2019). Nevertheless, further diagnostic efforts are needed
to properly understand the process.

## 4    Conclusions

In this paper, we investigated the thermodynamic structure of atmospheric stratification over the Namibia coast and the adjacent
SEA Ocean, to understand the formation of stratocumulus cloud decks and the dispersion of aerosol particles and generally
pollutants at the regional scale. Using a combination of ERA5 reanalysis and observations by satellite sensors (COSMIC/GPS-



RO and CATS), we investigated the temporal (diurnal/seasonal) and spatial (meridional/zonal) variability of the frequency, intensity and depth of atmospheric inversion layers, in which the following findings were deduced.

The combined presence of the South Atlantic high (SAh) pressure system and the baroclinity region set up by land-ocean waves constitutes the primary driver of low-level inversion over the region. Low-level inversion is predominant over the coast and the ocean than on land due to the core of subsidence occurring in the marine environment. In the inland sub-region, inversion and SAh are significantly influenced by surface radiative heating and cooling.

Surface-based (SB) and elevated-based (EL) inversion types are often linked to distinct spatial-temporal characteristics. Meanwhile, the base height associated with the discrimination of the types attains its peak during summer and is lowest in winter.

Diurnal inversions are nocturnal in behaviour over the region and predominantly occur in winter, coinciding with the peaks in depth and strength, a condition influenced mainly by subsidence intensity and differences in surface heating and cooling. Typically, inversion frequency exceeds 85 times in winter, peaking around May and often occurs between 20 and 40 times in summer. Spatially, these parameters are higher over the ocean, especially at the coast, than over land.

Aerosol transported over the inversion is mainly trapped and completely cut off from interaction with lower clouds, such as the stratocumulus cloud decks beneath the inversion, thereby hindering their continuous growth. Nevertheless, these particulates, which are temporarily stagnant aloft the inversion layer or being recirculated to the subcontinent, are often removed by scavenging high-level clouds.

Evidence shows that the formation of stratocumulus cloud decks commonly seen over the SEA Ocean and the adjacent Namibia coast is significantly aided by low-level inversion. Thus, the absence of inversion mostly corresponds to no cloud decks at the lower troposphere. The presence, regardless of magnitude, will always create the formation of low-level cloud decks.

Model output from ERA5 demonstrated strong agreement with GPS-RO observations following the correlation between the two exceeding 0.95, and the general uncertainty has a minimal value of ±0.014.

*Data availability.* COSMIC GPS-RO data and ERA5 data were obtained from the COSMIC Data Analysis and Archive Center (CDAAC), available at https://cdaac-www.cosmic.ucar.edu/cdaac/products.html and https://cds.climate.copernicus.eu/datasets respectively. The CATs data is available from https://cats.gsfc.nasa.gov/data.

*Author contributions.* ATY, DK and HH analysed the data with contributions by RB and SJP. ATY, DK wrote the paper with contributions from HH, SJP, RB and PF.



*Competing interests.* The authors have no competing interests to declare.

*Acknowledgements.* The authors appreciate the Copernicus Climate Change Service (C3S) Climate Data Store (CDS) for the use of the ERA5 product. We thank the Cosmic Data Analysis and Archive Center (CDAAC) for the GPS-RO data and NASA Goddard Space Flight

390 Center (GSFC) for the CATS product. This research was funded by the North-West University (NWU) Unit for Environmental Sciences and Management (UESM). We acknowledge the support of the National Research Foundation of South Africa (NRF) under grant numbers.: 120764, 106431, 129320, and 114691. P. Formenti acknowledges financial support by the French National Research Agency under grant agreement no 662ANR-15-CE01-0014-01, the French national program LEFE/INSU, the French National Agency for Space Studies (CNES) as part of the AErosol RadiatiOn and CLOuds in southern Africa (AEROCLO-sA) project.



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



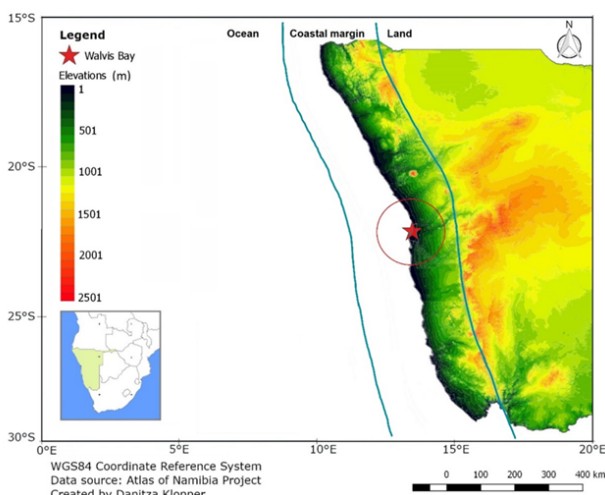

**Figure 1.** Relief map of Namibia over the area of interest. Walvis Bay is indicated by the red star (22°58'43.5" S 14°38'28.4" E, 97 m above mean sea level) and is circled by a 100 km radius. This greater area is divided into three smaller regions, namely, the ocean (to the west), the coastal margin (outlined in blue), and the land (to the east).





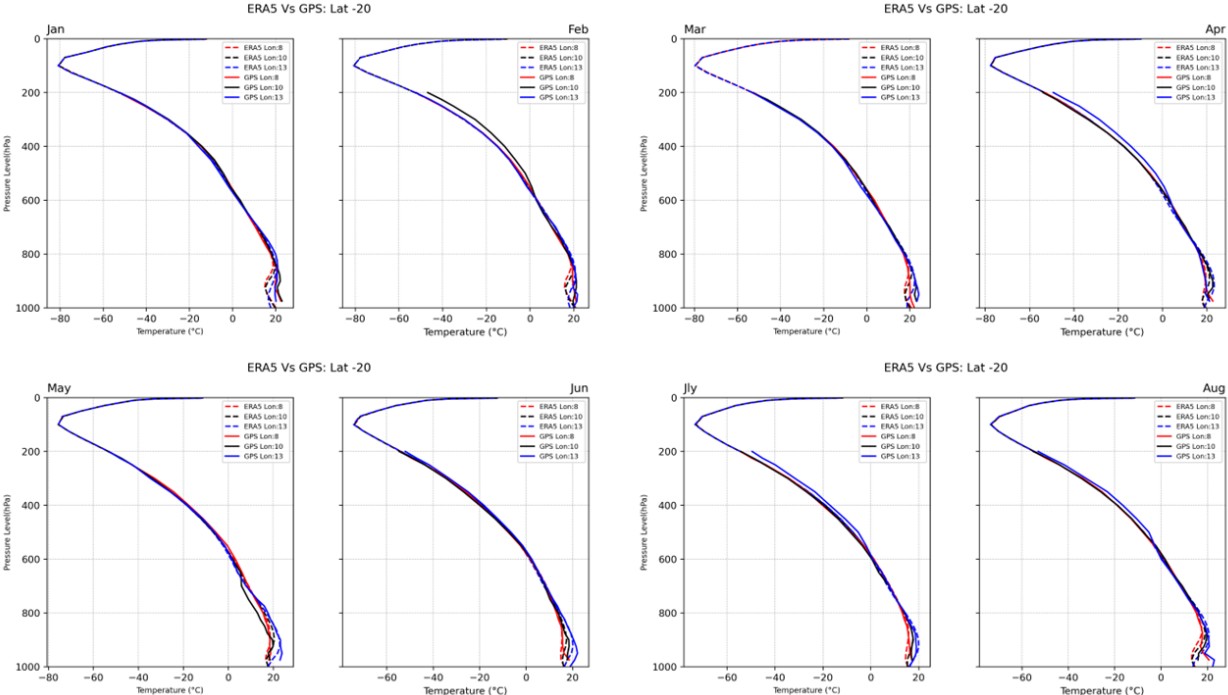

**Figure 2.** Comparison between ERA5 model and GPS-RO datasets for January to August 2007-2017 over the spatial range of latitude 20° S and longitude 8-13° E.





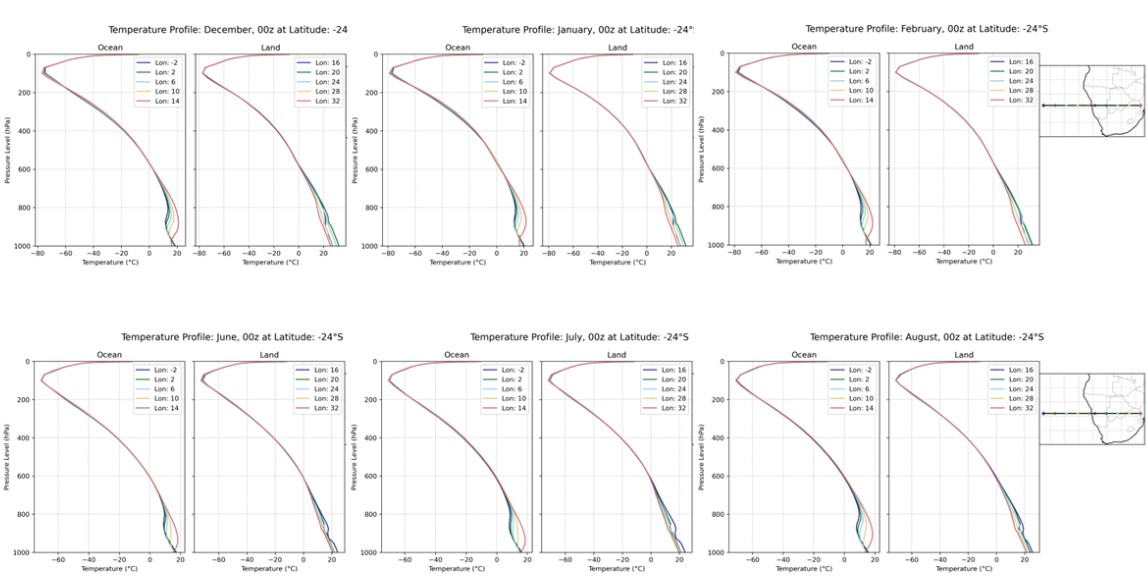

**Figure 3.** Surface inversion over SEA Ocean and adjacent Namibia at 00:00 UTC for DJF and JJA from the ERA5 product.



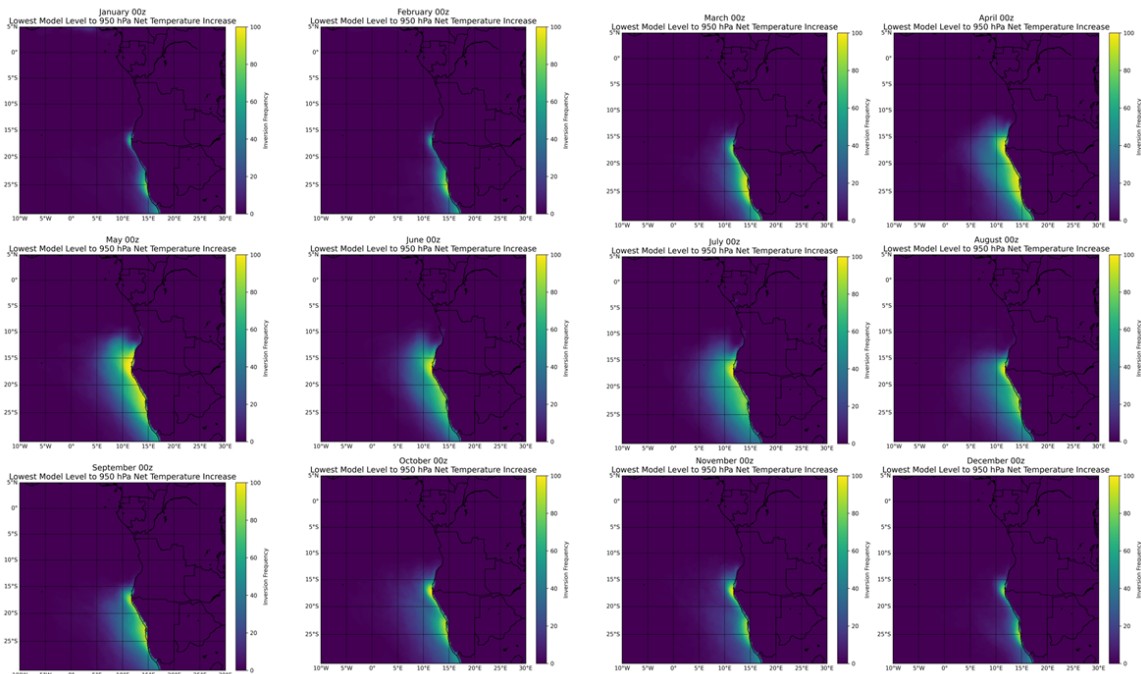

**Figure 4.** ERA5 monthly averages of inversion frequency between the surface and 950 hPa at 00:00 UTC.





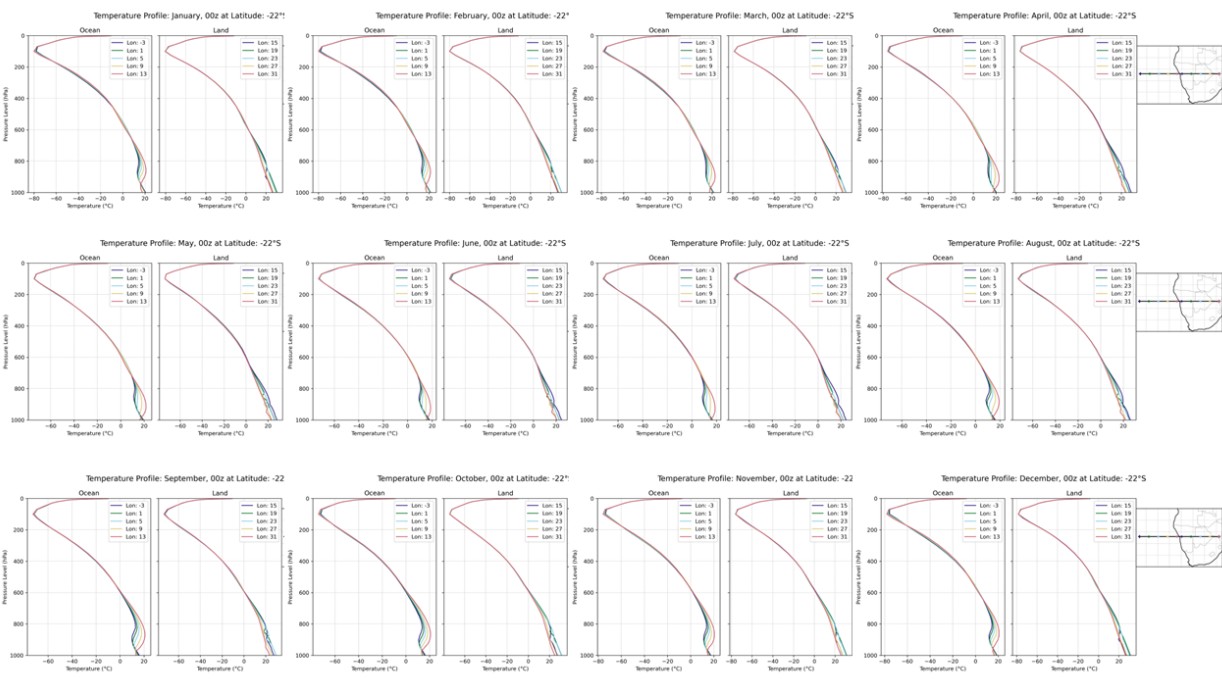

**Figure 5.** Plots of the inversions illustrate the monthly average variation in inversion strength and depth between January and December at 00:00 UTC for locations 22° S and 3 to 31° E from ERA5 reanalysis data.





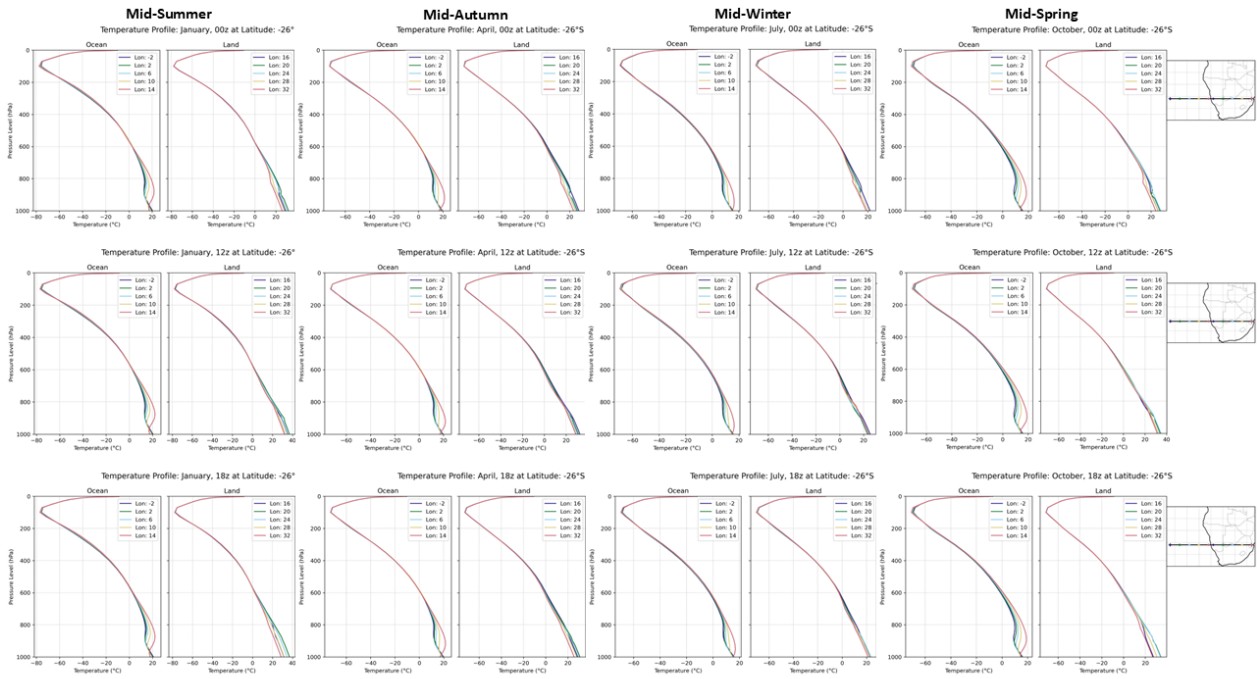

**Figure 6.** Diurnal variation of the inversion in the timesteps 00:00, 12:00 and 18:00 UTC for the months January (mid-summer), April (mid-autumn), July (mid-winter) and August (mid-spring) at locations 26° S and -2° to 32° E from the ERA5 product.



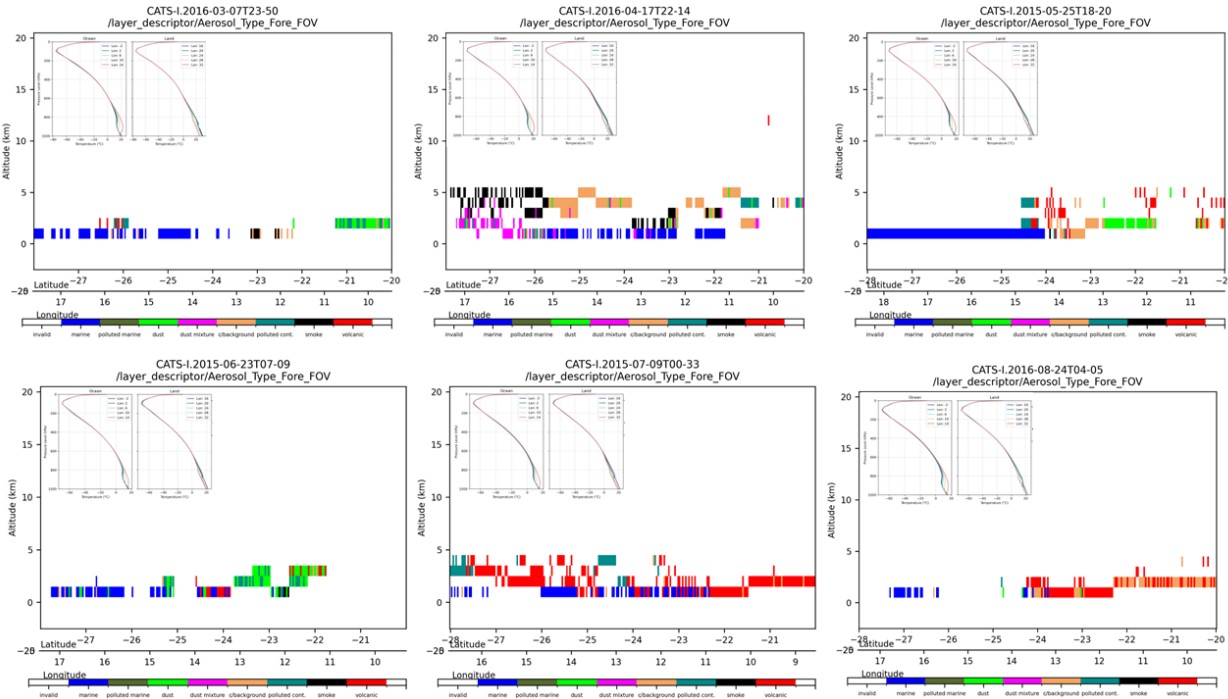

**Figure 7.** Aerosol transport observation from CATS data compared with collocated monthly average inversion plots from the ERA5 product(insertion) for autumn and winter days (March to August) between 2015 and 2017 at latitude 26° S.





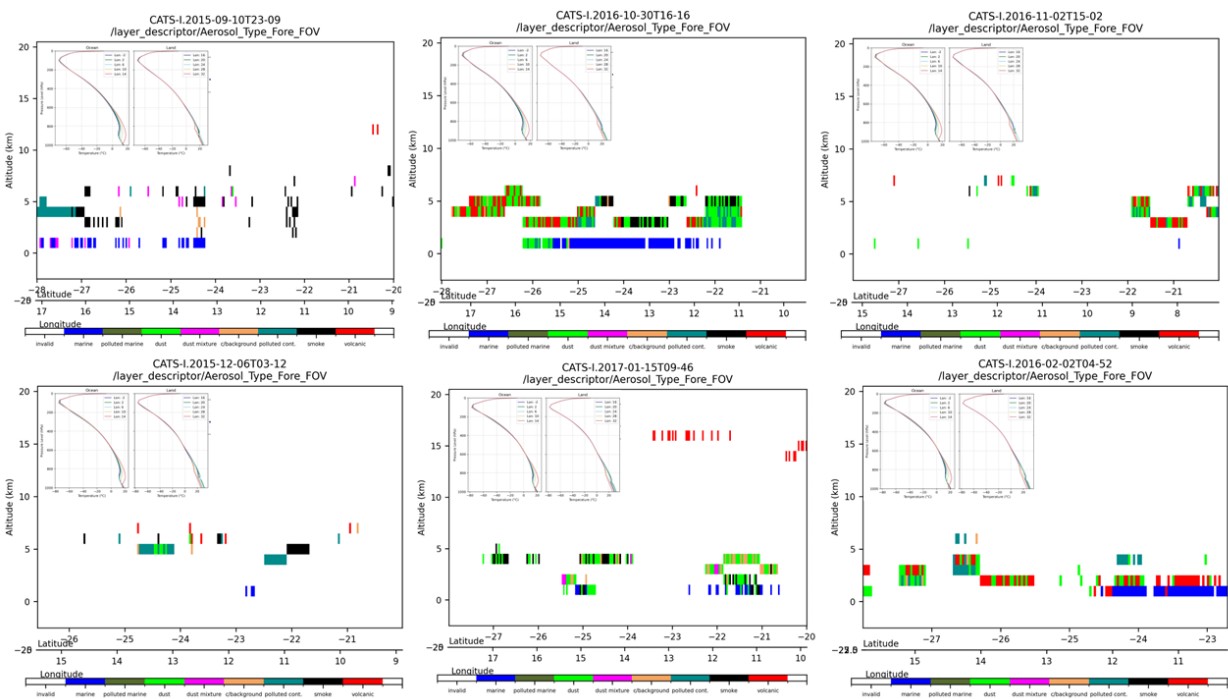

**Figure 8.** The same as above but for spring and summer days (September to February) between 2015 and 2017 at latitude 26° S.





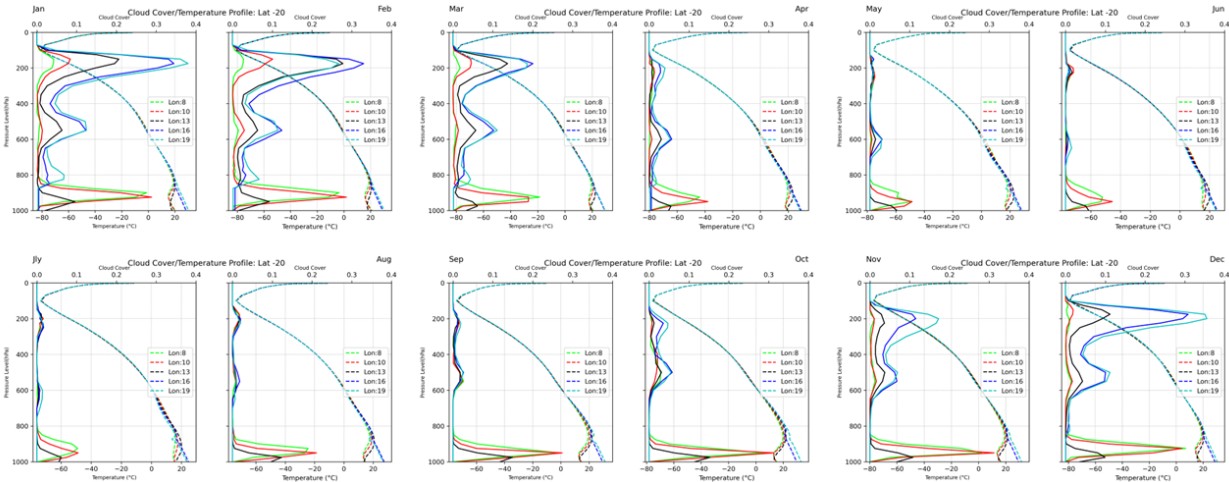

**Figure 9.** Combined plot of temperature profile and cloud cover over latitude 20° S and longitude range 8-19° E for the monthly averages between 2007 and 2017.



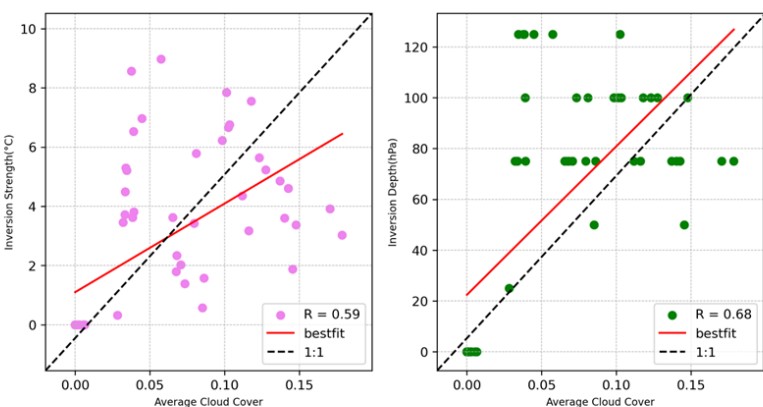

**Figure 10.** Comparison between (a) inversion strength and average cloud cover and (b) inversion depth and cloud cover for 2007-2017.