# Peer review of "Atmospheric stratification over the southeast Atlantic Ocean adjacent to the Namibian coast"

_EGUsphere, 2025_

## Referee Comment (RC1)

**Review of the manuscript „Atmospheric stratification over the southeast Atlantic Ocean adjacent to the Namibian coast" by Yakubu et al.**

In their manuscript „Atmospheric stratification over the southeast Atlantic Ocean adjacent to the Namibian coast", Yakubu et al. utilize 11 years of ERA5 reanalysis data together with global positioning system radio occultation (GPS-RO) observations to analyze the frequency and characteristics of temperature inversions along the Namibian coastline. For a number of cases, space-borne lidar observations from the cloud-aerosol-transport system (CATS) are used to link the characteristics of the inversion to the vertical distribution of aerosols. Connections between the inversion and low-level clouds are analyzed on the basis of ERA5 data. The authors find that inversions are much more frequent offshore and along the coast rather than inland and the occurrence peaks during winter when the inversion base height is also lowest.

The topic is relevant and appropriate for publication in ACP, and the observational data is in principal of high interest considering this is a region of sparse observational data in general. However, unfortunately, I find the paper to be poorly written, structured, and not well prepared in general (a figure seems to be missing, some units are missing, figures are unclear). The analysis remains superficial and is not well presented, many statements are not based on results nor are references provided. One example for the lacking quality of this manuscript: In the last paragraph of the introduction, goals of the analysis are defined, but are never analyzed or even mentioned again in the manuscript (see major point 1). Questions remain about the data selection: Why is monthly data used when daily is available and monthly data will clearly lead to biases in inversion strength estimates by mixing situations when the inversion is present and not present, and cannot really be used for the case studies presented using the CATS data? Why are 1km vertical resolution data from CATS used to analyze the connection between the inversion and aerosols above, when frequently, these are known to be within a few 100 meters, and when data with a 60m vertical resolution are available? Uncertainties of the data used are not discussed adequately: CATS classifies a substantial fraction of the aerosols as volcanic in about half the cases but this is never mentioned or discussed. GPS-RO and ERA5 data disagree substantially where it matters most (near the inversion of the boundary layer), but the differences are not discussed and characterized in detail, nor are any conclusions drawn from this.

On this basis I cannot recommend the manuscript to be published in ACP.

**Major points:**
1) In the last paragraph of the introduction, the authors set the stage for what the paper will present in the following way (italics are a direct quote of parts of the last paragraph of the introduction): „*This study uses an 11-year climatology (2007–2017) of thermal atmospheric stratification below 10 km above ground level (AGL) over Namibia and the adjacent Atlantic Ocean, combining ERA5 reanalysis model data with GPS-RO observations from the COSMIC mission to evaluate the regional atmospheric stratification's spatial and temporal variability. It seeks to understand the impact of stratus clouds on the formation of the stratocumulus cloud deck and to investigate the relationship between the strength of atmospheric stability, as identified by the data, and the lower tropospheric stability (LTS), which has been shown to correlate with cloud fraction in the region.*" I have highlighted two statements that I will comment on: Blue: This is the only mention of stratus clouds in the entire paper - there is no context given for this statement in the introduction, there is no analysis to this end in the paper, and no concluding statement. Red: This is the only time that LTS is mentioned in the entire manuscript. It is never calculated, there is no analysis regarding this, and there is no concluding statement.
2) Lack of references: In some (not all) parts of the manuscript, there is speculation on the attribution of patterns to physical processes. In many cases, there is no evidence presented for these speculations, and often there are no references provided (see minor comments).
3) Section 3.1:
    1) Questions about the data: GPS-RO: no data available below 500 meters, however, from the plots it looks like only the lowest 25hPa are not shown - this should correspond to about 200 meters? Please elaborate further
    2) L130: I agree with the authors that the data sets agree remarkably well everywhere in many atmospheric levels. However, this is not the case between 800 and 1000 hPa. This is

clearly relevant as these are the pressure levels where the inversion typically sits to cap the boundary layer. I am not convinced by the strong correlation provided in L137 (r > 0.95), as this is clearly driven by the (very) good agreement of the two data sets above the capping inversion of the PBL. In my opinion, it would be much more meaningful to show correlations of the estimated inversion height/depth/strength between the two data sets. Also, I am questioning why the authors decided to use monthly mean data for the analysis. Using monthly mean data should cause biases in the estimated inversion characteristics, as this mixes situations with and without an inversion being present (e.g. leading to an inversion strength that is biased low).

   3) In my opinion, it would be helpful to show the actual distributions of the inversion characteristics height/strength/depth from the two data sets and correlate these, in the best case for daily data.

4) Section 3.2
   1) L158: SB inversion is defined as an inversion between 0 and 50 m AGL, but how are these detected when the data has a vertical resolution of 100m (GPS) or at least 200m (ERA5)? It would be better if the authors clearly defined SB by criteria that can be represented with their data.
   2) L160: The results described here are hard to see in the figures, as always the full profiles are shown. It would be much easier to follow if e.g. the authors just showed the seasonality/diurnal cycle and spatial gradients of the inversion characteristics in one dedicated plot each.
   3) L181: The authors are using reanalysis data at the levels 1000, 975 and 950 hPa to detect inversions, and argue that none are detected inland - how should they considering these pressure levels are subsurface there?

5) Section 3.3
   1) In principal, connecting the inversion characteristics to aerosols and clouds is relevant. However, I believe the data selection to be a poor choice. CATS profiles are shown, which are available at 60m resolution, but are shown at 1km vertical resolution. In the top of each overpass panel, the monthly average temperature profile from reanalysis is shown (pixelated and not readable), which is then described as a „collocation". Again, the choice to use monthly mean data for the reanalysis seems to limit the information that one can derive, for individual cases. Regarding the CATS profiles, how can 1km vertical resolution be useful, when often times, the distance between the aerosol and cloud (and therefore inversion) layers in the SEA much less than that (Rajapakshe et al. 2017, Gupta et al. 2021)?
   2) CATS seems to frequently classify aerosols as volcanic, which does not seem reasonable in this region, however, this is neither discussed, nor is there any uncertainty/quality discussion on the data.

6) Section 3.4
   1) L300-303: This is an interesting point to make, and indeed the seasonality between the SB inversion seems to agree with the occurrence of fog (directly at the coast: The paper by Nagel is for Swakopmund). This topic justifies a closer look, though, as other locations in the Namib feature a different fog seasonality (that is related to the EL inversions), peaking in spring and summer (Lancaster et al. 1984, Spirig et al. 2019, Andersen et al. 2019).
   2) Fig. 9: It is unclear what precisely is shown here, what is meant with average cloud cover over the region (which region?), and which locations are used for the inversion characteristics?
   3) Looking at the scatter plot, I am surprised to see that the two variables are still fairly strongly correlated (r = 0.68 and 0.59). In particular inversion strength and cloud cover seem to be anti-correlated when removing the data points that do not feature an inversion. Is this true, and how many data points are this?

7) Structure of the manuscript: I was confused about the sectioning of combining results and discussion in Sec. 3, then in each of the sections 3.1 to 3.4 results are described and discussed (so far ok), but then Sec. 3.5 is named Discussion. This entire section is speculative, and provides very little references to other literature.

**Minor points:**
• L18: I am surprised to see the authors state that Namibia partially lies in the mid-latitudes when its southernmost point is < 29°S

- L28-45: This part of the introduction is not easy to understand and follow and needs to be restructured.
- L80: The vertical resolution of the ERA5 levels data needs to be described.
- L105-106: „The spatial and temporal distribution of valid profiles across the study area is given in Fig. 1." This figure seems to be missing from the manuscript.
- L108: I guess this refers to the GPS data, as the reanalysis has a vertical resolution of 25hPa?
- L120: Information on the vertical resolution of the data set is missing but relevant.
- Fig. 1: The bar scale of the map seems to be wrong (at 23°S a 1°x 1° box has length scales of around 100km, but the 5° coastal margin corresponds to less than 400km with the bar scale)
- L133 The authors mention EL inversion and SB inversion here, however, it is not described how these are defined and the term SB has not been established. I see that it is discussed further down in section 3.2, but it would need earlier explanation at the very least.
- L138/139 The uncertainty numbers provided are not clear - what are the units here?
- L164-168: For the described causal links neither results nor references are provided.
- L185: unit missing, also in the related figure.
- L188-193: References missing
- L194: Unit missing
- L195: „Like observed across the maritime environs" please correct this
- L204-212: References missing
- L230: Reference missing, in my opinion, most studies point to fog being advective
- L234/235: The „other associated factors" seem purely speculative and no analysis regarding these is presented.
- L242-252: References missing

References
- Andersen, H., Cermak, J., Solodovnik, I., Lelli, L., and Vogt, R.: Spatiotemporal dynamics of fog and low clouds in the Namib unveiled with ground- and space-based observations, Atmos. Chem. Phys., 19, 4383–4392, https://doi.org/10.5194/acp-194383-2019, 2019.
- Gupta, S., McFarquhar, G. M., O'Brien, J. R., Delene, D. J., Poellot, M. R., Dobracki, A., Podolske, J. R., Redemann, J., LeBlanc, S. E., Segal-Rozenhaimer, M., and Pistone, K.: Impact of the variability in vertical separation between biomass burning aerosols and marine stratocumulus on cloud microphysical properties over the Southeast Atlantic, Atmos. Chem. Phys., 21, 4615–4635, https://doi.org/10.5194/acp-21-4615-2021, 2021.
- Rajapakshe, C., Z. Zhang, J. E. Yorks, H. Yu, Q. Tan, K. Meyer, S. Platnick, and D. M. Winker (2017), Seasonally transported aerosol layers over southeast Atlantic are closer to underlying clouds than previously reported, Geophys. Res. Lett., 44, 5818–5825, doi:10.1002/2017GL073559.
- Seely, M. K. and Henschel, J. R.: The Climatology of Namib Fog, Proceedings, First International Conference on Fog and Fog Collection, Vancouver, Canada, 353–356, 1998.
- Spirig, R., Vogt, R., Larsen, J. A., Feigenwinter, C., Wicki, A., Parlow, E., Adler, B., Kalthoff, N., Cermak, J., Andersen, H., Fuchs, J., Bott, A., Hacker, M., Wagner, N., Maggs-Kölling, G., Wassenaar, T., and Seely, M.: Probing the fog life-cycles in the Namib desert, B. Am. Meteorol. Soc., https://doi.org/10.1175/BAMSD-18-0142.1, 2019.